# Longitudinal Analysis of Social Isolation and Cognitive Functioning among Hispanic Older Adults with Sensory Impairments

**DOI:** 10.3390/ijerph20156456

**Published:** 2023-07-27

**Authors:** Corinna Trujillo Tanner, Jeremy Yorgason, Avalon White, Chresten Armstrong, Antonia Cash, Rebekah Case, Joshua R. Ehrlich

**Affiliations:** 1College of Nursing, Brigham Young University, Provo, UT 84602, USA; jeremy.yorgason@byu.edu (J.Y.);; 2Department of Ophthalmology and Visual Science, University of Michigan Medical School, Ann Arbor, MI 48109, USA

**Keywords:** vision impairment, hearing impairment, dual sensory impairment, hispanic, older adult, cognition

## Abstract

Objectives: Understanding the intersection of age, ethnicity, and disability will become increasingly important as the global population ages and becomes more diverse. By 2060, Hispanics will comprise 28% of the U.S. population. This study examines critical associations between sensory impairment, social isolation, and cognitive functioning among Hispanic older adults. Methods: Our sample consisted of 557 Hispanic older adults that participated in Rounds 1–3 or Rounds 5–7 of the National Health and Aging Trends Study. Longitudinal mediation models across a three-year span were estimated using Mplus, with vision, hearing, and dual sensory impairments predicting cognitive functioning directly and indirectly through social isolation. Results: Findings indicated that cognitive functioning was concurrently and, in certain cases, longitudinally predicted by vision and dual sensory impairments and by social isolation. Contrary to expectations, vision and hearing impairments were not predictive of social isolation. Dual sensory impairment was associated with social isolation, yet no significant indirect associations were found for sensory impairments predicting cognitive functioning through social isolation. Discussion: The finding that social isolation did not mediate the relationship between sensory impairment and cognitive decline among Hispanic older adults in the U.S. is contrary to findings from other studies that were not specifically focused on this population. This finding may be evidence that culturally motivated family support and intergenerational living buffer the impact of sensory impairments in later life. Findings suggest that Hispanic older adults experiencing dual sensory impairments may benefit from interventions that foster social support and include family members.

## 1. Introduction

Consistent with population trends in the United States (U.S.), the median age of Hispanics is increasing, but at a more rapid pace than in the general population [1]. Hispanics are the largest ethnic group in the U.S., comprising 60.6 million individuals and accounting for 18% of the population. Demographic trends predict that by the year 2060, Hispanics will comprise 28% of the U.S. population (111 million individuals) [2].

While vision loss affects 9% of the U.S. population, health disparities increase the risk and impact of vision loss for Hispanics [3]. Hispanics are approximately twice as likely to have diabetes and are the most likely racial-ethnic group in the U.S. to have undiagnosed and untreated hypertension, both of which are causes of retinopathy [4,5]. By 2050, half the people living with glaucoma will be Hispanic [6]. These disparities seem to be especially true for Hispanic women [7].

Hispanics may lack access to vision screening despite being at increased risk for vision loss. Brown estimated that 75% of Hispanic older adults living with glaucoma were undiagnosed and untreated [8]. Similarly, 63% of the participants in the Los Angeles Latino Eye Study who had vision impairments had never been diagnosed or sought treatment prior to the study [9].

When the need for refractive correction is unmet, vision impairment is compounded. While up to 64% of Hispanics over the age of 40 require glasses or contact lenses, 20% of those lack access, especially those with lower rates of acculturation, a lower education level, and who are uninsured [10].

Hearing loss affects 27% of adults 60–69 and 63.1% of adults 70 and older [11]. Although the prevalence of hearing loss among Hispanic and non-Hispanic samples is similar [12], those of Hispanic ethnicity will spend a greater proportion of their lives hearing impaired [13]. The impact of hearing impairment among Hispanics is further compounded by the underuse of hearing aids, which is correlated with a lack of health insurance [14].

Dual sensory impairment (DSI) consists of concurrent vision and hearing loss. Approximately 1.5 million adults in the U.S. aged 20 years and older have DSI, with only 1% of those being under age 70 [15]. In one longitudinal study, Hispanic participants reported a higher proportion of dual sensory impairment [16]. Findings indicate that those with DSI are more likely to experience cognitive decline and report loneliness [17]. Additionally, one international study has shown that DSI can be associated with higher social disconnectedness and increased loneliness in certain populations [18]. Social isolation has been associated with an increased risk of dementia [19].

Sensory impairments (VI, HI, and DSI) are each independently associated with an increased risk of cognitive decline [20,21,22]. These relationships are complex, and various mechanisms have been hypothesized [21]. Yet social isolation, commonly associated with sensory impairments, could partially explain this relationship [19]. Sensory impairments increase the risk of social isolation by limiting social participation and impacting psychosocial health [23,24]. Social isolation is associated with a myriad of negative health outcomes, including the risk of depression and subsequent cognitive decline [25]. This association has been shown to significantly accelerate brain aging and contribute to earlier death [26]. Longitudinal, population-based research has identified social isolation as a mechanism linking sensory impairment with cognitive functioning among a national sample of older adults with sensory impairments in the U.S. [27].

While sensory impairments generally increase the risk of social isolation, certain cultural factors may buffer this impact. Hispanics are twice as likely to live in intergenerational households compared to non-Hispanic whites [28]. This may be due to the cultural value of familism. The term *familism* is defined as a reliance on and prioritization of family relationships and describes “a cultural frame of reference about the centrality of the family that is enacted in attitudes and behaviors” [29] (p. 464). Many cultures and ethnicities around the world may be considered familistic [30]. Terms that describe this primacy of the family include communalism in African cultures, filial piety in Asian Cultures, and familism among Hispanic cultures [30]. Hispanics may prefer to receive social support from family rather than from community resources [31,32]. Past research has identified a link between the cultural value of familism and perceived social support as well as psychological well-being, social well-being, and reduced distress [30,33]. Campos et al. described how there was a sense of closeness and social support among those who were more familistic, which buffered negative psychological outcomes [33]. Although these benefits were experienced by individuals from European, African, and Hispanic cultures, those who were Hispanic had higher levels of familism [33]. Although some research has explored how social isolation differs among older adults based on ethnic origins [34], little is known about the direct influence of culture. The buffering effect of familism among Hispanics may partly explain the cultural health advantage described by the *Hispanic Paradox* [35,36]. The Hispanic Paradox describes a phenomenon that describes how, despite having fewer financial resources, less access to education, and higher health disparities, Hispanic individuals in the U.S. tend to live longer [35].

Although Hispanics are a highly diverse group of individuals of various religions, socio-economic backgrounds, and education levels, persistent health disparities, including less access to healthcare, lower rates of insurance, higher poverty rates, and discrimination, may play a role in long-term health outcomes [13,14,37,38]. Among Hispanics over the age of 65, 14% have a diagnosis of Alzheimer’s disease [39]. While Hispanics are 1.5 times more likely to experience dementia compared with non-Hispanic whites [40], the reasons are not well understood. Minority groups often experience delayed diagnosis and inadequate treatment of dementia [41]. Additionally, misunderstandings among Hispanics about what constitutes normal aging and issues of trust between minority groups and the medical establishment further compound these issues [41].

Increased ethnic diversity and rapidly shifting demographics present a need for research on the unique challenges faced by Hispanic older adults to inform health care, policy, and funding priorities. We hypothesized that among Hispanic older adults, having one or more self-reported sensory impairments would be associated with poorer cognitive functioning cross-sectionally and longitudinally across 1 and 2 years, and that these relationships would be associated indirectly through social isolation.

## 2. Methods

Our sample consisted of 557 older Hispanic adults that participated in the National Health and Aging Trends Study (NHATS), a nationally representative study of Medicare beneficiaries in the United States aged 65 and older. The current study combined Hispanic participants from Rounds 1, 2, and 3 (*n* = 342) and Rounds 5, 6, and 7 (*n* = 215) to create a total sample size of 557 Hispanic participants measured across three years. We only retained participants from Rounds 5–7 who were not included in the sample from Rounds 1–3, and so all 557 participants were unique.

### 2.1. Measures

#### 2.1.1. Cognitive Functioning

We used cognitive measures collected in NHATS corresponding to orientation, executive function, and learning/memory [42]. To measure orientation, respondents were asked to recite the date, president, and vice president of the U.S. Higher scores represent higher levels of cognitive orientation. Executive function was assessed using a clock-drawing test. Respondents were given a blank piece of paper and asked to draw a clock with hands placed at 11:10. Drawings were scored ranging from 0 (not recognizable as a clock) to 5 (accurate depiction) (specific criteria are found in the NHATS User Guide, pages 88–89). Higher scores indicated better executive function. Learning/memory was evaluated using a delayed word recall test. Interviewers read a list of 10 words to respondents, who were then asked to recall as many of the words as possible. After a 5 min delay, respondents were again asked to recall as many of the words as possible. Higher scores for the delayed word recall measure indicated better learning/memory.

#### 2.1.2. Sensory Impairment

Three separate variables were used to assess self-reported vision impairment (VI), hearing impairment (HI), and dual sensory impairment (DSI). VI was measured using a total of three items. If the participant reported being blind or that they were unable to see well enough (including when using corrective lenses—glasses or contact lenses) to recognize someone across the street or to read newspaper print, they were then coded as having a VI. This method was used in prior studies using NHATS data [43].

A dichotomous measure of HI was constructed using four items. If the respondent reported difficulty with any of the items, they were then coded as having a hearing impairment. Hearing impairment questions related to whether respondents could “hear well enough to carry on a conversation in a quiet room,” “hear well enough to carry on a conversation in a room with a radio or TV playing,” and “hear well enough to use the telephone”, and an item assessing whether participants were deaf (“yes” coded as 1, “no” coded as 0). Individuals were characterized as having HI only if hearing problems were severe enough to impact their functioning (whether or not they wore a hearing aid). People with hearing aids who did not report problems with these listed items were not coded as having a hearing impairment for this study.

Self-reported DSI was indicated if the participant reported having both HI and VI. In the current sample, 22% (*n* = 114) reported HI, 15% (*n* = 85) reported VI, and 7% (*n* = 37) reported DSI.

#### 2.1.3. Social Isolation

We measured social isolation with the goal of identifying elements of familial and other support that might be more relevant to Hispanic individuals [32]. Social isolation scores were calculated by aggregating each of the following possibilities, with “yes” coded as 1 and “no” coded as 0 for each of the following: (a) living alone; (b) having one or fewer people who he/she talked to in the last year about important things; (c) not attending religious services; (d) not visiting friends/family in the participant’s home or in the home of the friend/family member; and (e) not living in an intergenerational household. The resulting measure had a scale ranging from zero to five, with higher scores indicating greater social isolation.

#### 2.1.4. Covariates

We included conceptually relevant covariates, and descriptive information about each of these constructs can be found in Table 1. Covariates included chronological age, biological sex (labeled gender), marital status, education, smoking status, and self-rated health. Education, rather than income, was included as an indicator of socioeconomic status [44] because of its association with cognitive functioning [45] and strong correlation with income.

### 2.2. Analysis

Longitudinal models that examined indirect associations between sensory impairment, social isolation, and cognitive functioning across a three-year span were estimated using Mplus. We first estimated models with sensory impairments predicting cognitive functioning, with social isolation modeled as a mediator in that relationship. We then added covariates. More specifically, Model 1 included separate models for vision and hearing impairments or dual sensory impairments, predicting each measure of cognitive functioning directly and indirectly through social isolation (see Figure 1). Model 2 included each of those models with covariates added, including age, gender, marital status, education, smoking status, and illness comorbidities. Full information maximum likelihood was used to address missing data. We used bootstrapping with 5000 draws to adjust the standard errors of indirect associations. 

## 3. Results

### 3.1. Preliminary Analyses

As seen in Table 1, our sample of Hispanic older adults had an age range of 65 to greater than 90 and consisted of about 57% females. The majority of our sample (77%) had at least a high school diploma, and almost half had at least some college education. The average income was a little over USD 56,000 per year. Correlations between the main study variables were in expected directions and of anticipated magnitudes (see Table 2).

### 3.2. Direct Associations

How do sensory impairments relate to social isolation concurrently and across time? Contrary to our expectations, neither VI nor HI were significantly associated with social isolation concurrently or across time. Unlike VI and HI alone, DSI was associated with higher social isolation one year later in both Model 1 (*β* = 0.079, *p* < 0.05) and Model 2 (*β* = 0.086, *p* < 0.05).How do sensory impairments relate to cognitive functioning across time among Hispanic older adults? As seen in Model 1 of Table 3, VI was negatively associated with concurrent orientation (*β* = −0.111, *p* < 0.01) and executive function scores (*β* = −0.109, *p* < 0.05), as well as with learning/memory (*β* = −0.124, *p* < 0.05) and executive function (*β* = −0.103, *p* < 0.01) one year later. VI was also associated with lower executive function scores one year later when covariates were added in Model 2 (*β* = −0.079, *p* < 0.05). HI was not associated with measures of cognitive functioning in this sample. As seen in Table 4, DSI was negatively associated with orientation scores concurrently in Model 1 (*β* = −0.094, *p* < 0.05) and with learning/memory across one year in Model 1 (*β* = −0.143, *p* < 0.01) and Model 2 (*β* = −0.091, *p* < 0.05).

3.What is the relationship between social isolation and cognitive functioning among Hispanic older adults? As displayed in Table 3, social isolation was negatively associated with concurrent orientation in Model 2 (*β* = −0.091, *p* < 0.05) and executive function in Model 1 (*β* = −0.099, *p* < 0.05) and Model 2 (*β* = −0.113, *p* < 0.05). Social isolation was also related to lower learning/memory across one year in Model 1 (*β* = −0.084, *p* < 0.05) and Model 2 (*β* = −0.111, *p* < 0.01).

### 3.3. Indirect Associations

4.Are sensory impairments related indirectly to cognitive functioning through social isolation? There were no significant indirect associations between VI, HI, or DSI and cognitive functioning through social isolation.

In summary, as anticipated, VI and DSI were associated with cognitive functioning, but unexpectedly, HI was not. Social isolation was associated with lower cognitive functioning. However, aside from DSI, individual sensory impairments were not related to social isolation in this Hispanic sample. Between 30% and 43% of the variance (as per r-squared) in cognitive functioning measures was accounted for in the statistical models.

## 4. Discussion

Given the increasingly diverse U.S. population, this study examined associations between sensory impairments, social isolation, and cognitive functioning among Hispanics, who comprise the largest ethnic group in the US. We hypothesized that having one or more sensory impairments would be associated with poorer cognitive functioning cross-sectionally and longitudinally across 1 and 2 years, and that these relationships would be mediated by social isolation.

### 4.1. Summary of Findings

We began by exploring how sensory impairments relate to social isolation and found that VI and HI were not related to social isolation concurrently or across 1 or 2 years. This finding was unexpected, as VI and HI have historically been shown to be associated with social isolation [23,24]. The lack of correlation between sensory impairments and social isolation in this study may suggest an effect of cultural modification. Hispanics are twice as likely to live in intergenerational households compared to their non-Hispanic white peers [28] and are known to rely on family for many types of support [31]. Familism is not unique to Hispanics and is described as filial piety in Asian cultures and communalism in African cultures [30]. In addition to social benefits, familism is associated with better physical and psychological outcomes [46]. High levels of familism are related to less loneliness [47]. While the term social isolation tends to describe quantitative measures of social opportunities, as in the present study, the term loneliness often describes the subjective experience or feeling of being socially isolated. High levels of familism have also been associated with improved physical health symptoms and lower levels of depression [47].

DSI was associated with social isolation longitudinally, but not cross-sectionally. One interpretation of this finding is that DSI does not have an immediate impact on social isolation, but over time it may start to impact connection with others in important ways [48]. Social isolation can have profound negative impacts on older individuals’ (ages 65+) health and is associated with increased risk for anxiety, depression, and dementia [26,49].

We also explored how sensory impairments relate to cognitive functioning across time among Hispanic older adults. The present study found that VI was associated with concurrent and longitudinal impacts on cognitive function. This is consistent with meta-analyses that have linked VI with cognitive decline and clinically diagnosed dementia (e.g., [19]). Cao et al. [50] report similar findings longitudinally as well as cross-sectionally, regardless of subjective or objective measures of VI.

Despite abundant research linking HI to cognitive decline [19,49], we did not find an association between HI and cognitive functioning. There may be certain protective factors, such as familism, that buffer the impacts of hearing loss on cognitive decline among Hispanics [36]. Another possible explanation for the different findings is variation in the definition and measurement of vision loss across studies.

We explored the relationship between social isolation and cognitive functioning among Hispanic older adults with sensory impairments. Social isolation was related to lower orientation and executive functioning concurrently and to lower learning/memory one year later. Social isolation, among other potentially modifiable risk factors, is associated with 40% of dementia cases globally [49]. The Lancet Commission on Dementia cites social isolation as a key risk factor for dementia, with social isolation having a significant population attributable fraction (PAF) of 3.5% [19].

Finally, we explored whether sensory impairments indirectly relate to cognitive functioning through social isolation. No indirect associations were significant for VI or HI, predicting cognitive functioning through social isolation. Past research has identified social isolation as an important mediator of the relationship between sensory impairments and cognitive functioning [27]. This non-finding may, again, be evidence of an effect modification by culture, owing to familism.

Examples from various cultures demonstrate the buffering power of family ties. The term filial piety describes family supports that honor parents, grandparents, and ancestors in societies influenced by Confucian thought [51]. Research surrounding filial piety suggests that intergenerational support, including emotional and financial support, is effective in improving the overall health of older adults, including physical and psychological well-being, performance of activities of daily living (ADL), instrumental activities of daily living (IADL), and later life cognition [52]. While support from family has been described as being more static and obligatory, it can also be described as being more reliable [53]. This can be increasingly important in older age as mobility restrictions and chronic conditions such as sensory impairments increase. Regardless of sociodemographic background or gender, Ying et al. [54] found that more extended family ties were associated with better cognition. Familism found among Hispanic populations can create a barrier between stress and adversity, and negative health outcomes [55]. Fifty-seven percent of the sample in the current study lived in intergenerational households. The buffering effect of this family support may partly explain the unexpected lack of association between sensory impairments and social isolation in this study.

### 4.2. Strengths and Limitations

This research fills a gap in the current research landscape by exploring the intersection of ethnicity, aging, and disability in a national sample of Hispanic older adults with sensory impairments. Understanding the influence of culture and the dynamics impacting family caregiving should inform health programs and the delegation and prioritization of resources. To our knowledge, this is the first paper to explore social isolation accruing from sensory impairments among a uniquely Hispanic sample.

Due to the smaller sample size, these analyses had lower power than could be achieved with a larger sample size.

### 4.3. Future Directions

Beginning in 2022, future rounds of the NHATS data will oversample Hispanic participants to achieve a proportion more consistent with what is seen in the general population. NHATS data currently include self-reported sensory impairment. Future rounds of NHATS data (beginning with data collection in 2021) will include objective measures of both vision and hearing. This may allow for analyses that focus on the spectrum of severity of sensory impairment as it relates to long-term outcomes. Future studies may compare samples from the U.S. with samples from outside the U.S., such as those in the Mexican Health and Aging Study (MHAS). We recommend future research on the qualitative nature of family support among Hispanic families as it relates to living with sensory loss. Future measures of social isolation could include measures of social activities that are relevant to those of Hispanic ethnicity, e.g., going on picnics, attending dances, visiting family, etc.

## 5. Conclusions

Familism buffers stress, improves psychosocial outcomes, and is associated with higher self-esteem and subjective wellbeing in times of stress [30,47]. In the present study, social isolation was not imminent among those experiencing sensory impairments when family support was present. Our findings suggest that familism may play a significant role in health trajectories among Hispanic older adults living with sensory impairments.

Culture is an important part of the context in which health events occur. As we improve our understanding of the dynamic relationship between culture, aging, and disability, we are better able to develop targeted interventions to support Hispanic Older adults towards positive outcomes.

## Figures and Tables

**Figure 1 ijerph-20-06456-f001:**
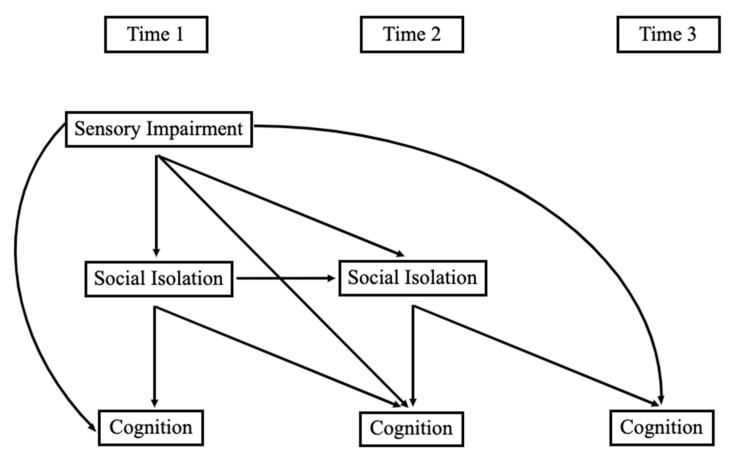
Conceptual model. Note: Time 1 was in 2011 or 2015; Time 2 was in 2012 or 2016; Time 3 was in 2013 or 2017.

**Table 1 ijerph-20-06456-t001:** Descriptive statistics from main study variables among Hispanic participants of NHATS (*n* = 557).

Variable	*n* (%) or Mean (SD)	Range
Age Groups		
65–69	151 (27.11%)	
70–74	141 (25.31%)	
75–79	106 (19.03%)	
80–84	88 (15.80%)	
85–89	46 (8.26%)	
≥90	25 (4.49%)	
Sex		
Male (coded as 1)	246 (44.17%)	
Female (coded as 0)	311 (55.83%)	
Education Level		
Less than High School	318 (57.09%)	
High School	87 (15.62%)	
Trade/Some college	83 (14.90%)	
College degree	69 (12.39%)	
Marital Status		
Married	276 (49.64%)	
Not Married	280 (50.36%)	
Intergenerational household	320 (54.45%)	
Smoking Status		
Smoker	19 (5.57%)	
Non-Smoker	322 (94.43%)	
Hearing Disability		
Hearing Loss	117 (21.47%)	
No Hearing Loss	417 (78.53%)	
Vision Disability		
Vision Loss	85 (15.29%)	
No Vision Loss	471 (84.71%)	
Dual Sensory Disability		
Hearing/Vision Loss	37 (6.98%)	
No Hearing/Vision Loss	493 (93.02%)	
Social Isolation		0–4
Time 1	1.88 (0.98)	
Time 2	1.81 (0.98)	
Executive Function		0–5
Time 1	3.19 (1.17)	
Time 2	3.22 (1.32)	
Time 3	3.10 (1.37)	
Learning/Memory		0–9
Time 1	3.01 (1.76)	
Time 2	2.95 (1.87)	
Time 3	2.83 (1.89)	
Orientation		0–8
Time 1	5.50 (1.42)	
Time 2	5.63 (1.58)	
Time 3	5.42 (1.62)	
Health	1.20 (0.89)	0–4

**Table 2 ijerph-20-06456-t002:** Correlations between main study variables among Hispanic participants of the NHATS study (*n* = 557).

	1	2	3	4	5	6	7	8	9	10	11	12	13
1. T1 HL	1												
2. T1 VL	0.25 ***	1											
3. T1 DI	0.52 **	0.65 **	1										
4. T1 Social Iso	0.04	−0.05	−0.05	1									
5. T2 Social Iso	0.06	0.00	0.06	0.56 ***	1								
6. T1 Exec Fun	−0.00	−0.08 *	−0.05	−0.08 *	−0.10 *	1							
7. T2 Exec Fun	−0.02	−0.15 **	−0.10 *	−0.13 **	−0.06	0.56 ***	1						
8. T3 Exec Fun	−0.04	−0.12 *	−0.06	−0.04	0.01	0.50 ***	0.56 ***	1					
9. T1 Learning	−0.07	−0.05	−0.04	0.01	0.04	0.14 **	0.16 ***	0.15 **	1				
10. T2 Learning	−0.11 *	−0.15 **	−0.17 ***	−0.05	−0.07	0.16 ***	0.29 ***	0.32 ***	0.44 ***	1			
11. T3 Learning	−0.12 *	−0.01	−0.08	−0.05	−0.06	0.18 ***	0.24 ***	0.35 ***	0.37 ***	0.57 ***	1		
12. T1 Orient	−0.09 *	−0.11 *	−0.07	−0.04	−0.01	0.20 ***	0.25 ***	0.23 ***	0.20 ***	0.25 ***	0.20 ***	1	
13. T2 Orient	−0.06	−0.12 **	−0.11 *	−0.01	−0.06	0.22 ***	0.29 ***	0.34 ***	0.21 ***	0.36 ***	0.32 ***	0.60 ***	1
14. T3 Orient	−0.10	−0.10	−0.11 *	0.04	0.01	0.28 ***	0.41 ***	0.39 ***	0.17 **	0.31 ***	0.36 ***	0.41 ***	0.49 ***

Notes: HL = Hearing Loss. VL = Vision Loss. DI = Dual Sensory Impairment (Hearing and Vision Loss). Social Iso = Social Isolation. Exec Fun = Executive Function. Orient = Orientation. * *p* < 0.05, ** *p* < 0.01, *** *p* < 0.001.

**Table 3 ijerph-20-06456-t003:** Standardized regression coefficients from the structural equation model of vision and hearing impairment predicting cognitive functioning indirectly through social isolation.

	Learning/Memory B (SE)		Orientation B (SE)		Executive Function B (SE)	
Predictor Variables	Model 1	Model 2	Model 1	Model 2	Model 1	Model 2
VI -> T1 CF	−0.035 (0.04)	0.009 (0.04)	−0.111 ** (0.04)	−0.045 (0.04)	−0.109 * (0.05)	−0.056 (0.04)
VI -> T2 CF	−0.124 * (0.05)	−0.077 (0.05)	−0.045 (0.04)	−0.020 (0.04)	−0.103 ** (0.04)	−0.079 * (0.04)
VI -> T3 CF	0.073 (0.05)	0.093 † (0.05)	−0.046 (0.04)	−0.029 (0.04)	−0.057 (0.05)	−0.046 (0.05)
HI -> T1 CF	−0.070 (0.05)	−0.022 (0.05)	−0.076 † (0.05)	−0.030 (0.04)	0.022 (0.05)	0.065 (0.04)
HI -> T2 CF	−0.037 (0.05)	−0.001 (0.05)	−0.003 (0.04)	0.020 (0.04)	0.004 (0.04)	0.035 (0.04)
HI -> T3 CF	−0.080† (0.04)	−0.056 (0.04)	−0.035 (0.05)	−0.017 (0.04)	−0.031 (0.05)	−0.012 (0.05)
T1 Social Isolation -> T1 CF	0.009 (0.04)	0.008 (0.04)	−0.045 (0.04)	−0.091 * (0.04)	−0.099 * (0.04)	−0.113 * (0.05)
T2 Social Isolation -> T2 CF	−0.084 * (0.04)	−0.111 ** (0.04)	−0.047 (0.04)	−0.037 (0.04)	0.002 (0.04)	−0.042 (0.04)
T2 Social Isolation -> T3 CF	−0.010 (0.04)	−0.037 (0.04)	0.047 (0.05)	−0.017 (0.05)	0.078 † (0.04)	0.084 † (0.05)
VI -> T1 Social Isolation	−0.066 (0.05)	−0.058 (0.05)	−0.066 (0.05)	−0.056 (0.05)	−0.064 (0.05)	−0.055 (0.05)
VI -> T2 Social Isolation	0.009 (0.04)	0.022 (0.04)	0.009 (0.04)	0.022 (0.04)	0.009 (0.04)	0.022 (0.04)
HI -> T1 Social Isolation	0.056 (0.05)	0.049 (0.05)	0.057 (0.05)	0.051 (0.05)	0.055 (0.05)	0.048 (0.05)
HI -> T2 Social Isolation	0.044 (0.04)	0.051 (0.04)	0.043 (0.04)	0.050 (0.04)	0.043 (0.04)	0.050 (0.04)
VI -> T1 Social Isolation -> T1 CF	−0.001 (0.00)	0.000 (0.00)	0.003 (0.00)	0.005 (0.01)	0.006 (0.01)	0.006 (0.01)
VI -> T2 Social Isolation -> T2 CF	−0.001 (0.00)	−0.002 (0.01)	0.000 (0.00)	−0.001 (0.00)	0.000 (0.00)	−0.001 (0.00)
VI -> T2 Social Isolation -> T3 CF	0.000 (0.00)	−0.001 (0.00)	0.000 (0.00)	0.000 (0.00)	0.001 (0.00)	0.002 (0.00)
HI -> T1 Social Isolation -> T1 CF	0.001 (0.00)	0.000 (0.00)	−0.003 (0.00)	−0.005 (0.01)	−0.005 (0.01)	−0.005 (0.01)
HI -> T2 Social Isolation -> T2 CF	−0.004 (0.01)	−0.006 (0.01)	−0.002 (0.00)	−0.002 (0.00)	0.000 (0.00)	−0.002 (0.00)
HI -> T2 Social Isolation -> T3 CF	0.000 (0.00)	−0.002 (0.00)	0.002 (0.00)	−0.001 (0.00)	0.003 (0.00)	0.004 (0.01)
Sample Size (N)	557	557	557	557	557	557
Chi-Square Model Fit	0.70	0.27	10.19	10.81	70.90*	100.19 *
RMSEA	0.00	0.00	0.00	0.00	0.05	0.07
CFI	10.00	10.00	10.00	10.00	0.99	0.99
R^2^ of T3 CF	0.35 ***	0.43 ***	0.30 ***	0.38 ***	0.37 ***	0.40 ***

Notes: † *p* < 0.01, * *p* < 0.05, ** *p* < 0.01, *** *p* < 0.001. VI = Vision Impairment; HI = Hearing Impairment; CF = Cognitive Functioning; RMSEA = Root Mean Squared Error of Approximation; CFI = Comparative Fit Index; T = Time, where T1 was in 2011 or 2015, T2 was in 2012 or 2016, and T3 was in 2013 or 2017; Arrows (dash and greater than sign) denote a directional regression, and multiple arrows denote an indirect effect relationship. Model 2 included covariates of age, education, biological sex, health, marital status, and smoking status.

**Table 4 ijerph-20-06456-t004:** Standardized regression coefficients from the structural equation model of dual sensory impairment predicting cognitive functioning indirectly through social isolation.

	Learning/Memory B (SE)		Orientation B (SE)		Executive Function B (SE)	
Predictor Variables	Model 1	Model 2	Model 1	Model 2	Model 1	Model 2
DSI -> T1 CF	−0.041 (0.04)	0.013 (0.04)	−0.094 * (0.04)	−0.031 (0.04)	−0.064 (0.05)	−0.015 (0.05)
DSI -> T2 CF	−0.143 ** (0.04)	−0.091 * (0.04)	−0.060 (0.05)	−0.026 (0.05)	−0.081 † (0.04)	−0.042 (0.04)
DSI -> T3 CF	−0.001 (0.04)	0.013 (0.04)	−0.050 (0.05)	−0.031 (0.05)	−0.017 (0.05)	0.000 (0.05)
T1 Social Isolation -> T1 CF	0.006 (0.04)	0.008 (0.04)	−0.047 (0.04)	−0.091 * (0.04)	−0.097 * (0.04)	−0.110 * (0.05)
T2 Social Isolation -> T2 CF	−0.077 * (0.04)	−0.106 * (0.04)	−0.044 (0.04)	−0.034 (0.04)	0.008 (0.04)	−0.038 (0.04)
T2 Social Isolation -> T3 CF	−0.012 (0.04)	−0.039 (0.04)	0.048 (0.05)	−0.017 (0.05)	0.077 † (0.04)	0.083 † (0.05)
DSI -> T1 Social Isolation	−0.05 (0.05)	−0.056 (0.05)	−0.052 (0.05)	−0.055 (0.05)	−0.051 (0.05)	−0.056 (0.05)
DSI -> T2 Social Isolation	0.079 * (0.04)	0.086 * (0.04)	0.079 * (0.04)	0.086 * (0.04)	0.079 * (0.04)	0.086 * (0.04)
DSI -> T1 Social Isolation -> T1 CF	0.000 (0.00)	0.000 (0.00)	0.002 (0.00)	0.005 (0.01)	0.005 (0.01)	0.006 (0.01)
DSI -> T2 Social Isolation -> T2 CF	−0.006 (0.01)	−0.009 (0.01)	−0.003 (0.00)	−0.003 (0.00)	0.001 (0.00)	−0.003 (0.00)
DSI -> T2 Social Isolation -> T3 CF	−0.001 (0.00)	−0.003 (0.00)	0.004 (0.00)	−0.001 (0.01)	0.006 (0.00)	0.007 (0.01)
Sample Size (N)	557	557	557	557	557	557
Chi-Square	0.90	0.27	10.05	10.80	70.94 *	100.14 *
RMSEA	0.00	0.00	0.00	0.00	0.05	0.07
CFI	10.00	10.00	10.00	10.00	0.99	0.99
R^2^ of T3 CF	0.34 ***	0.42 ***	0.30 ***	0.38 ***	0.36 ***	0.40 ***

Notes: † *p* < 0.01, * *p* < 0.05, ** *p* < 0.01, *** *p* < 0.001. DSI = Dual Sensory Impairment; CF = Cognitive Functioning; RMSEA = Root Mean Squared Error of Approximation; CFI = Comparative Fit Index; T = Time, where T1 was in 2011 or 2015, T2 was in 2012 or 2016, and T3 was in 2013 or 2017; Arrows (dash and greater than sign) denote a directional regression, and multiple arrows denote an indirect effect relationship. Model 2 included covariates of age, education, biological sex, health, marital status, and smoking status.

## Data Availability

The data used were from the National Health and Aging Trends Study (NHATS) and can be accessed at: https://nhats.org/.

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
