# Peer review of "Longitudinal Analysis of Social Isolation and Cognitive Functioning among Hispanic Older Adults with Sensory Impairments"

_ijerph, 2023, doi:10.3390/ijerph20156456_

Round 1
Reviewer 1 Report
The authors propose a study that examines critical associations between sensory impairment, social isolation and cognitive functioning among Hispanic older adults. Below I will point out some of the aspects that could be improved to achieve a higher quality of the manuscript.
- Given that the title focuses on loneliness and its consequences, perhaps lines 78-83 could be expanded further.
- Line 128-137 a more concise description of the subjects could be included.
- Line 298-302: needs to include references and comparisons with other studies where this is not the case.
- Line 300 include a reference is needed
- Line 306 needs to be referenced
- The conclusions should be improved for a better understanding of the results and the manuscript.
I consider it to be a correct and well-written manuscript in which the sections are well arranged. Some of the references should be added but in general the changes to be made are minor.
Reviewer 2 Report
This study is unique in that it presents results that differ from studies conducted with other cultures and races.
However, it needs to be revised before publication, and the revisions are as follows:
1) There is redundancy in the introduction. The introduction of the meta-analysis findings needs to be combined with the sensory impairment content that precedes it. Overall, the introduction needs to be condensed.
2) The characterization of the results of this study is described as familism. Therefore, it is necessary to compare the results of this study with studies conducted in other cultures that exhibit familism. Also, the effect of familism on health outcomes needs further consideration.
Reviewer 3 Report
This article offers a robust examination of the connections between sensory impairments, social isolation, and cognitive functioning among Hispanic older adults in the US, a demographic that is often underrepresented in such studies. It interestingly revealed that contrary to prior research, sensory impairments, specifically vision impairment (VI) and hearing impairment (HI), were not related to social isolation. This deviation is attributed to the cultural concept of familism common in Hispanic communities, which may provide a support network that mitigates social isolation. However, the study also found that dual sensory impairments (DSI) could lead to social isolation over time, underscoring the need for interventions in such cases.
The authors' assertion that VI has both concurrent and longitudinal impacts on cognitive function is well-supported, aligning with previous research. However, their conclusion that HI does not impact cognitive functioning among the Hispanic population seems to contradict prevailing literature. This divergence is attributed to potential cultural protective factors, such as familism, but it warrants further investigation.
A noteworthy limitation of this research is the smaller sample size, reducing the power of the analyses. The study also raises questions about the definition and measurement of vision and hearing loss, suggesting the need for a more standardized approach across studies. It is commendable that the authors address the influence of culture on health outcomes and emphasize the importance of considering culture when developing interventions for specific populations.
While the study effectively fills a research gap and provides a fresh perspective, it also signals the need for more comprehensive, larger-scale studies, especially those incorporating objective measurements of sensory impairments. Incorporating culturally relevant measures of social isolation and a deeper exploration of the qualitative nature of family support could further enrich future research in this area.
Authors in the Introduction should integrate prior findings as this study stands in broader scientific context, and it substantiates the importance of investigating the potential influences of sensory impairments on cognitive function, while also considering social and psychological factors like depression. Authors should allude to the fact that previous research has identified the relationship between cognitive impairment and depression in older adults (doi: 10.3390/ijerph20075290, 10.3389/fnagi.2023.1121190, 10.1017/S0033291712000402). Specifically, it has been found that cognitive performance deficits are significantly associated with clinically significant depression symptoms, functional disability and mortality in older US adults. The former addition would strengthen the argument for a comprehensive approach to understanding cognitive function in the context of sensory impairments and other potentially influential factors.
From a proofreading perspective, the article is relatively sound with appropriate language usage and sentence structures. However, there are a few areas where clarity and coherence could be improved. Some sentences are quite long and complex, potentially obscuring the intended meaning. For example, the sentence "Unlike VI and HI alone, DSI was associated with social isolation, but only across 1 year and beyond, not cross sectionally" could be split into two sentences for easier readability. Similarly, using clearer subheadings within the "Discussion" section could make the information more digestible.
From a scientific perspective, the introduction of a more standardized measure for sensory impairment, which the authors themselves suggested, would significantly improve the quality and generalizability of the research. The paper should also consider more nuanced investigation into the types and degrees of sensory impairments rather than treating VI and HI as homogeneous categories.
Future research should aim to replicate these findings in larger and more diverse Hispanic populations, accounting for variations across subgroups such as differences in country of origin, acculturation levels, and socioeconomic status. As the authors suggested, there is also a need for future rounds of NHATS data to include objective measures of vision and hearing, allowing a clearer picture of how severity of sensory impairment relates to social isolation and cognitive functioning.
Given the central role that familism appears to play in mitigating the effects of sensory impairments, future research should delve deeper into this concept. It would be beneficial to explore the qualitative nature of family support and how it interacts with sensory loss in the Hispanic population. It would also be worthwhile to compare these findings with populations outside the U.S, such as those in the Mexican Health and Aging Study (MHAS), to better understand cultural factors at play. Additionally, the use of culturally relevant measures of social isolation (e.g., attending cultural dances, family visits) would add nuance to the understanding of this phenomenon within the context of the Hispanic community.
Minor editing required.
Reviewer 4 Report
GENERAL COMMENTS
Interesting topic, interesting results, well written. Acceptable for publication after minor change.
This reviewer is not familiar enough with the statistical methods employed to provide a judgment on their appropriateness.
SPECIFIC ITEMS
line 67f: The authors quote evidence that 1.5 million adults have DSI, then proceed to state that 11.3% of 80+ year olds experience DSI. For the sake of easier comparison, the authors should please the percentages and the absolute numbers for both populations.
